# Associations between vigorous physical activity, social ties, social support, and self-reported health among older adults in Accra, Ghana

**Nestor Asiamah**[1,2]*, **Kyriakos Kouveliotis**[3], **Richard Eduafo**[2], **Richard Borkey**[2]

**1** Division of Interdisciplinary Research and Practice, School of Health and Social Care, University of Essex, Colchester, United Kingdom, **2** Department of Geriatrics and Gerontology, Africa Centre for Epidemiology, Accra, Ghana, **3** Department of Health Care Management, International Telematic University Uninettuno, Corso Vittorio Emanuele II, Rome, Italy

* n.asiamah@essex.ac.uk, nestor.asiamah@ace-gh.org

**Data Availability Statement:** The data has been provided as supplementary information and an SPSS file.

## Abstract

Meeting recommended vigorous physical activity (VPA) levels represents a hallmark for healthy living, but VPA in older populations is likely to lead to casualties that may compel older adults to underscore their health. This study examined the associations between VPA, social ties, social support, and self-reported health in an African sample of older adults. This study adopted the cross-sectional design. The study population was community-dwelling older adults aged 60 years or higher in Accra, Ghana. A total of 686 older adults responded to self-reported questionnaires. Data were analysed with the Pearson's chi-square test and binary logistic regression. The study found that older adults who had above 5 children were 3 times (AOR = 3.169; p = .002) more likely to participate in VPA for 30 minutes or more a day compared with their peers without children. Having social support from between 1 and 5 people was 28 times (AOR = 28.215; p = .000) more likely to result in good health compared to not having anyone to source social support from. Older adults who participated in VPA for 30 minutes or more were less likely (AOR = 0.129; p = 0.000) to report good health compared with those who participated in VPA for less than 30 minutes. We conclude that when social ties and other personal characteristics are adjusted for, prolonged VPA does not necessarily enhance self-reported health, and not all social ties contribute to VPA and self-reported health.

## Introduction

Research to date shows that meeting recommended physical activity (PA) levels protects the individual against cardiovascular disease (e.g., stroke, hypertension, type 2 diabetes, etc.), depression, neurodegenerative diseases (i.e., dementia, Parkinson's disease), and other morbidities [1–3]. PA is therefore not only a potential medicine for older adults but is also a requirement for ageing well. Vigorous physical activity (VPA), however, highly engages the

**Funding:** The authors received no specific funding for this work.

**Competing interests:** The authors have declared that no competing interests exist.

body's bone and muscle mass and requires a higher energy expenditure relative to moderate PA. For this reason, intense PA can lead to dislocations, fractures, body pains, and related side effects in the general and older populations, and some researchers [1, 4] have acknowledged the possibility of VPA resulting in the above casualties in older people.

As researchers, we have had the opportunity to engage in informal dialogue with older adults in Accra as well as other African regions and heard many of them talk about their effort to modify their lifestyle and regularly participate in VPA; however, their daily engagement in prolonged VPA (i.e., vigorous activities lasting 30 minutes or more) resulted in dislocations, body pains, and other causalities. Some of these older persons added that they perceived their health to be badly affected by their uptake of routine VPA [5]. Our experience made us consider the possibility of VPA leading to poor self-reported health in older populations in harmony with the observation of some researchers [1, 5]. In the current study, we are driven by our curiosity to investigate the association between VPA and self-reported health, with social ties and demographic variables adjusted for. Serious implications for public health and health promotion are imperative if VPA leads to poor health in older populations after adjusting for relevant socio-demographic covariates, and an objective of this study is to identify and delineate these implications.

A portfolio of theories that includes the Social Capital Theory of Health (SCToH), Activity Theory of Ageing (ATA), and Continuity Theory of Ageing (CTA) recognises social connections as resources for maintaining PA and health. Research to date has also shown that health and wellbeing improve with increasing access to social capital in the general and older populations [6–9], albeit anecdotal and empirical evidence suggests that people with anti-social behaviours including social isolation abound in these populations, discourage health-seeking behaviour (e.g., PA) and, therefore, negatively affect health [8]. It has therefore become necessary for public health stakeholders, including social gerontologists, to promote population health by guiding older adults to savour social ties (to maintain PA and health) and avoid people who may have an unfavourable influence on them. This effort no doubt has an important place in healthy and active ageing programs and initiatives and ought to be made sustainable in all jurisdictions.

It is common knowledge that cultural and socio-economic status are factors that influence lifestyle (such as regular participation in VPA) and health. In developing countries, for instance, extreme poverty is high [3, 10] and may discourage peripheral social network members such as workmates from rendering financial support to their older relatives. Nevertheless, close friends may render non-financial support and encouragements that may enable older adults to meet recommended VPA levels in line with active ageing initiatives and maintain health. Moreover, grandparenting is common in developing countries such as Ghana and older guardians may be led by their grandchildren to access health care and health information. Similarly, grandchildren and children are likely to provide supports that can contribute to health in older populations, but they are unlikely to rub shoulders with their parents and grandparents in an African environment in routine vigorous physical activities (i.e., jogging, dancing, and cycling) that research [11–13] has confirmed as predictors of healthy ageing. Based on these explanations, we contend that different segments of peripheral social connections would relate to VPA and health differently.

We have observed in the literature that most empirical studies [6–9, 14], especially those conducted in Africa [8], measured and analysed social capital or ties concerning PA and self-reported health as a construct, which made it impossible for these studies to indicate the associations between different domains of social ties, VPA, and self-reported health. This situation makes it difficult for researchers and health promoters to understand the detail of the association between social ties and each of VPA and self-reported health to appraise the

aforementioned theories and spot possibilities for modifying and adapting them. We would want to recall that the SCToH recognises social ties and capital as determinants of health, but not all domains of social resources may contribute to health in specific socio-economic settings. We therefore deem it necessary to investigate the associations between different domains of social ties, VPA, and self-reported health to advance current debate in the area, identify potential implications for health promotion, and provide a basis for future longitudinal studies.

## Methods

### Study design

This study employed the cross-sectional design.

### Sample and selection

This study's population was community-dwelling older adults aged 60 years or higher in Accra, Ghana. The study setting was the head office of the Social Security and National Insurance Trust (SSNIT) in Accra where access to eligible older participants was guaranteed. Participants were volunteer older people who visited the said office in September and October 2019. The criteria for selecting participants were: (a) being aged 60 or higher; (b) having at least a basic educational qualification (that was an indicator of the ability to read and write in English, which was the medium in which questionnaires were administered); and (c) being free of all physical and mental disabilities (e.g., blindness, mental disorders) that precluded PA. At the premises of SSNIT, a physician assistant screened for eligible participants with a questionnaire. The physician assistant asked questions about the potential participant's age, current place of residence, educational level, and physical functional ability. Seven hundred and thirty (730) older adults met these inclusion criteria. We utilised the G*Power 3.1.9.4 software and relevant statistics (i.e., effect size 0.2, power = 0.8, α = 0.05)[5] for chi-square tests (or binary logistic regression) to calculate a minimum sample of 321 for the study. To maximise the generalisability of our findings, data were gathered on all 730 eligible participants.

### Variables and measurement

The main variables measured were social ties (the main predictors), VPA (i.e., first outcome variable) and self-reported health (i.e., second outcome variable). We measured VPA in terms of the time (in minutes) older adults spent in intense physical activities in a typical day following WHO's Global Physical Activity Questionnaire II. However, VPA was measured as a categorical variable made up of two groups (i.e., VPA lasting less than 30 minutes and VPA lasting 30 or more minutes), which enabled us to ascertain if prolonged VPA (per the views of older people we informally communicated with earlier) was associated with poor health. Self-reported health was thus measured as a dichotomous variable made of the levels 'poor health' and 'good health'. S1 Appendix is the survey containing all the measures whereas S2 Appendix shows the outcome variables, independent variables (social ties and social support), and control variables as well as information regarding their measurement and coding.

### Ethical considerations

This study received ethical approval from the ethics review committee of the Africa Centre for Epidemiology. The ethics review number was 001-2018-ACE. All participants provided written informed consent before agreeing to participate in the study voluntarily.

## Data gathering

This study received ethics approval from the ethics review committee of the Africa Centre for Epidemiology (ethics number– 001-2018-ACE). All the participants provided written informed consent before participating in this study. The informed consent form used provided details about the study's risks and benefits to respondents and society. The questionnaires were hand-delivered to participants in the study area between September 3 and October 31, 2019. To maximise the response rate, participants were politely asked to respond to questionnaires instantly. Out of 730 questionnaires administered, 697 were completed and returned; however, 11 of the questionnaires were not usable as they were filled half-way. So, 686 questionnaires were analysed.

## Statistical analysis method

Data were analysed using IBM SPSS Statistics, version 26 (IBM SPSS Inc., NY, USA). Descriptive statistics (i.e., frequency and percentage) were used to summarise the data. Missing items were handled following the technique employed by Bempong and Asiamah [15]. Findings were presented using binary logistic regression, but before using this statistical tool, the Pearson's chi-square test was used to assess group differences concerning self-reported health and VPA (see Table 1). At the level of logistic regression, associations were estimated in terms of Adjusted Odd Ratios (AORs) given the control variables incorporated into the analysis. Values of $p < .05$ were considered statistically significant. In the analysis, two independent logistic regression models were fitted, with the first model assessing the association between social ties and VPA. The second model evaluates the influence of VPA on self-reported health, with social ties and demographic variables captured as covariates. Variables included as potential covariates were variables previously reported as potential correlates of physical activity and health [3, 5, 7].

## Findings

In Table 1, 26% (n = 175) of all older adults engaged in VPA for less than 30 minutes a day, while 74% (n = 511) of all older adults engaged in VPA for at least 30 minutes a day. Moreover, 34% (n = 231) of all older people reported having poor health, whereas 66% (n = 455) of all older adults reported having good health. It can be seen from the table that all social ties and control variables were related to VPA at a 5% significance level. For example, 'Children' as a variable was significantly associated with VPA (Chi-square = 28.64; p < 0.001). All social ties and control variables were significantly associated with self-reported health at 5% significance level, except Dependents (Chi-square = 1.160; p = .560) and Age (Chi-square = 0.295; p = .587).

In Table 2, older adults who had more than 5 children were 3 (OR = 3.169; p < 0.05) times more likely to engage in VPA for 30 or more minutes compared with their peers who had no children. Older people who had between 1 and 5 siblings were less likely (OR = 0.017; p < 0.001) to engage in VPA for 30 or more minutes compared with those who did not have siblings. Older adults who had between 1 and 5 dependents (OR = 0.187, p < 0.05) and above 5 dependents (OR = 0.219, p < 0.05) were less likely to engage in VPA for 30 or more minutes compared with their peers who did not have any dependents. Elders who had between 1 and 5 close and mutual friends (OR = 0.045; p < 0.001) were less likely to participate in VPA for 30 or more minutes compared with those who did not have any close or mutual friends, and those who had more than 5 close and mutual friends were 3 times (OR = 3.00; p < 0.05) more likely to engage in VPA for 30 or more minutes compared with their peers who did not have any close or mutual friends. Older adults who had above 5 people to give them social support

**Table 1. Group differences by VPA and self-reported health.**

| Variable | Group | Total | VPA (minutes/day) | | Chi-square | p | Self-reported health | | Chi-square | p |
|---|---|---|---|---|---|---|---|---|---|---|
| | | n (%) | <30% | ≥30% | | | Poor % | Good % | | |
| Children | None | 49(7.1) | 4.1 | 3.1 | 28.64 | 0.000 | 6.1 | 1.0 | 69.994 | 0.000 |
| | 1–5 | 532(77.6) | 17.3 | 60.2 | | | 21.4 | 56.1 | | |
| | Above 5 | 105(15.3) | 4.1 | 11.2 | | | 6.1 | 9.2 | | |
| | Total | 686(100.0) | 25.5 | 74.5 | | | 33.7 | 66.3 | | |
| Siblings | None | 42(6.1) | 4.1 | 2.0 | 40.392 | 0.000 | 4.1 | 2.0 | 24.338 | 0.000 |
| | 1–5 | 385(56.1) | 12.2 | 43.9 | | | 16.3 | 39.8 | | |
| | Above 5 | 259(37.8) | 9.2 | 28.6 | | | 13.3 | 24.5 | | |
| | Total | 686(100.0) | 25.5 | 74.5 | | | 33.7 | 66.3 | | |
| Other blood relations | None | 21(3.1) | 2.0 | 1.0 | 19.574 | 0.000 | 2.0 | 1.0 | 22.292 | 0.000 |
| | 1–5 | 301(43.9) | 10.2 | 33.7 | | | 17.3 | 26.5 | | |
| | Above 5 | 364(53.1) | 13.3 | 39.8 | | | 14.3 | 38.8 | | |
| | Total | 686(100.0) | 25.5 | 74.5 | | | 33.7 | 66.3 | | |
| Dependents | None | 196(28.6) | 9.2 | 19.4 | 6.355 | 0.042 | 9.2 | 19.4 | 1.16 | 0.560 |
| | 1–5 | 399(58.2) | 13.3 | 44.9 | | | 19.4 | 38.8 | | |
| | Above 5 | 91(13.3) | 3.1 | 10.2 | | | 5.1 | 8.2 | | |
| | Total | 686(100.0) | 25.5 | 74.5 | | | 33.7 | 66.3 | | |
| Close and mutual friends | None | 28(4.1) | 3.1 | 1.0 | 37.708 | 0.000 | 3.1 | 1.0 | 65.488 | 0.000 |
| | 1–5 | 455(66.3) | 15.3 | 51.0 | | | 26.5 | 39.8 | | |
| | Above 5 | 205(29.6) | 7.1 | 22.4 | | | 4.1 | 25.5 | | |
| | Total | 686(100.0) | 25.5 | 74.5 | | | 33.7 | 66.3 | | |
| Social support | None | 77(11.2) | 4.1 | 7.1 | 25.757 | 0.000 | 3.1 | 8.2 | 10.024 | 0.007 |
| | 1–5 | 504(73.5) | 20.4 | 53.1 | | | 23.5 | 50.0 | | |
| | Above 5 | 105(15.3) | 1.0 | 14.3 | | | 7.1 | 8.2 | | |
| | Total | 686(100.0) | 25.5 | 74.5 | | | 33.7 | 66.3 | | |
| Gender | Male | 343(50.0) | 8.2 | 41.8 | 30.447 | 0.000 | 11.2 | 38.8 | 38.697 | 0.000 |
| | Female | 343(50.0) | 17.3 | 32.7 | | | 22.4 | 27.6 | | |
| | Total | 686(100.0) | 25.5 | 74.5 | | | 33.7 | 66.3 | | |
| Education | Basic | 147(21.4) | 11.2 | 10.2 | 74.151 | 0.000 | 6.1 | 15.3 | 13.944 | 0.001 |
| | Secondary | 49(7.1) | 2.0 | 5.1 | | | 4.1 | 3.1 | | |
| | Tertiary | 490(71.4) | 12.2 | 59.2 | | | 23.5 | 48.0 | | |
| | Total | 686(100.0) | 25.5 | 74.5 | | | 33.7 | 66.3 | | |
| Age (years) | 60–64 | 511(74.5) | 13.3 | 61.2 | 62.532 | 0.000 | 25.5 | 49.0 | 0.295 | 0.587 |
| | ≥64 | 175(25.5) | 12.2 | 13.3 | | | 8.2 | 17.3 | | |
| | Total | 686(100.0) | 25.5 | 74.5 | | | 33.7 | 66.3 | | |
| Income (₵) | ≤200 | 42(6.1) | 2.0 | 4.1 | 10.74 | 0.005 | 5.1 | 1.0 | 52.517 | 0.000 |
| | 201–400 | 70(10.2) | 4.1 | 6.1 | | | 4.1 | 6.1 | | |
| | ≥401 | 574(83.7) | 19.4 | 64.3 | | | 24.5 | 59.2 | | |
| | Total | 686(100.0) | 25.5 | 74.5 | | | 33.7 | 66.3 | | |
| Marital status | Not married | 413(60.2) | 21.4 | 38.8 | 55.544 | 0.000 | 25.5 | 34.7 | 35.165 | 0.000 |
| | Married | 273(39.8) | 4.1 | 35.7 | | | 8.2 | 31.6 | | |
| | Total | 686(100.0) | 25.5 | 74.5 | | | 33.7 | 66.3 | | |
| Employment status | Not employed | 217(31.6) | 11.2 | 20.4 | 16.615 | 0.000 | 15.3 | 16.3 | 30.766 | 0.000 |
| | Employed | 469(68.4) | 14.3 | 54.1 | | | 18.4 | 50.0 | | |
| | Total | 686(100.0) | 25.5 | 74.5 | | | 33.7 | 66.3 | | |

**Table 2. The association between social ties, social support, VPA, and self-reported health.**

| Variable | Group | VPA (minutes/day)[a] | | | | Self-reported health[b] | | | |
|---|---|---|---|---|---|---|---|---|---|
| | | OR | AOR | p | 95% CI | OR | AOR | p | 95% CI |
| Children | None[c] | 1 | 1 | - | - | 1 | 1 | - | - |
| | 01-May | 0.41 | 0.38 | 0.071 | ±0.96 | 0.05 | 0.04 | 0.000 | ±0.14 |
| | Above 5 | 3.19 | 3.17 | 0.002 | ±4.99 | 1.19 | 1.18 | 0.657 | ±1.89 |
| Siblings | None[c] | 1 | 1 | - | - | 1 | 1 | - | - |
| | 01-May | 0.05 | 0.02 | 0.000 | ±0.09 | 0.89 | 0.86 | 0.787 | ±2.33 |
| | Above 5 | 0.61 | 0.59 | 0.09 | ±0.77 | 5.58 | 5.64 | 0.000 | ±7.58 |
| Other blood relations | None[c] | 1 | 1 | - | - | 1 | 1 | - | - |
| | 01-May | 1.61 | 1.51 | 0.629 | ±7.81 | 0.33 | 0.34 | 0.166 | ±1.48 |
| | Above 5 | 1.58 | 1.55 | 0.178 | ±2.12 | 0.45 | 0.41 | 0.004 | ±0.53 |
| Dependents | None[c] | 1 | 1 | - | - | 1 | 1 | - | - |
| | 01-May | 0.21 | 0.19 | 0.001 | ±0.45 | 5.06 | 5.04 | 0.000 | ±10.33 |
| | Above 5 | 0.21 | 0.22 | 0.001 | ±0.47 | 5.15 | 5.12 | 0.000 | ±9.58 |
| Close and mutual friends | None[c] | 1 | 1 | - | - | 1 | 1 | - | - |
| | 01-May | 0.05 | 0.05 | 0.000 | ±0.15 | 0.10 | 0.12 | 0.002 | ±0.44 |
| | Above 5 | 3.03 | 3.00 | 0.001 | ±4.17 | 0.09 | 0.11 | 0.000 | ±0.18 |
| Social support | None[c] | 1 | 1 | - | - | 1 | 1 | - | - |
| | 01-May | 0.36 | 0.34 | 0.128 | ±1.29 | 28.27 | 28.22 | 0.000 | ±101.26 |
| | Above 5 | 0.19 | 0.17 | 0.006 | ±0.54 | 10.69 | 10.68 | 0.000 | ±17.49 |
| Gender | Male[c] | 1 | 1 | - | - | 1 | 1 | - | - |
| | Female | 2.61 | 2.62 | 0.003 | ±3.57 | 1.61 | 1.59 | 0.091 | ±1.79 |
| Education | Basic[c] | 1 | 1 | - | - | 1 | 1 | - | - |
| | Secondary | 0.08 | 0.05 | 0.000 | ±0.10 | 4.06 | 4.04 | 0.003 | ±8.64 |
| | Tertiary | 0.09 | 0.07 | 0.000 | 0.24 | 1.23 | 1.21 | 0.785 | ±4.44 |
| Age (years) | 55–64 [c] | 1 | 1.000 | - | - | 1 | 1 | - | - |
| | ≥65 | 5.69 | 5.62 | 0.000 | ±7.25 | 0.45 | 0.41 | 0.007 | ±0.57 |
| Income (₵) | ≤200[c] | 1 | 1 | - | - | 1 | 1 | - | - |
| | 201–400 | 11.01 | 10.95 | 0.000 | ±34.52 | 0.4 | 0.02 | 0.000 | ±0.08 |
| | ≥401 | 17.91 | 17.82 | 0.000 | ±61.17 | 1.43 | 1.40 | 0.568 | ±3.97 |
| Marital status | Not married[c] | 1 | 1 | - | - | 1 | 1 | - | - |
| | Married | 0.17 | 0.14 | 0.000 | ±0.20 | 0.51 | 0.49 | 0.006 | ±0.52 |
| Employment status | Not employed[c] | 1 | 1 | - | - | 1 | 1 | - | - |
| | Employed | 0.39 | 0.36 | 0.001 | ±0.46 | 0.28 | 0.25 | 0.000 | ±0.30 |
| VPA (minutes/day) | <30[c] | - | - | - | - | 1 | 1 | - | - |
| | ≥30 | - | - | - | - | 0.10 | 0.13 | 0.000 | ±0.21 |

[a]. Model 1: Nagelkerke = 0.545; [b]. Model 2: Nagelkerke = 0.565; [C]. Reference category; CI = confidence interval; OR = odd ratio (crude); AOR–adjusted odd ratio; p and 95% CI apply to only AOR.

were less likely (OR = 0.168; p < 0.05) to participate in VPA for 30 or more minutes compared to their counterparts who had no persons to give them social support. Older people who had between 1 and 5 dependents (OR = 5.041; p < 0.001) and above 5 dependents (OR = 5.121; p < 0.001) were 5 times more likely to report good health compared with their peers who did not have any dependents. Having social support from between 1 and 5 people is 28 times (OR = 28.215; p < .001) more likely to result in good health compared to not having anyone to source social support from. Similarly, older adults who had more than 5 people as a source of social support were 11 times (OR = 10.683; p < 0.001) more likely to report good health

compared with their colleagues who did not have any source of social support. Interestingly, older adults who participated in VPA for 30 minutes or more in a day were less likely (OR = 0.129; p < 0.001) to report good health compared with those who participated in VPA for less than 30 minutes per day.

## Discussion

This study assessed the associations between VPA, social ties, social support, and self-reported health in a Ghanaian sample of older adults aged 60 years or higher.

According to the study, having above 5 children is 3 times more likely to result in VPA lasting 30 or more minutes compared to having no children. This result supports the fact that about 68% of all older participants were employed (see Table 1) and thus possibly engaged in work-related PA, which Asiamah and Mensah [16] found in Ghana to be vigorous. Owing to the relatively low remunerations received by workers in Ghana and other African countries throughout the lifespan [8, 17, 18], individuals are compelled to maintain regular employment at old age to meet personal needs and those of dependents. Yet older people in Ghana and many other African countries retire after 60 and can as a result only work in industries (e.g., private security services, private commercial farming) that require employees to engage in work-related VPA [17]. Older persons who have 1 to 5 children are less likely to report good health compared to those who have no children. To explain, older adults may spend all or a greater part of their income to fend for their young children and dependents and may therefore lack the opportunity to expend their savings on personal health. From this perspective, the health benefits of vigorous PA older adults engaged in were possibly nullified by the financial and social burdens (and their accompanying health risks) associated with having up to 5 children and dependents. Considering the mixed pieces of evidence available in the literature regarding the above result [8, 13, 17, 19, 20], the current study implies that children who support the VPA of their older parents do not necessarily contribute to the self-reported health of these parents.

Older people having between 1 and 5 siblings are less likely to report VPA lasting 30 or more minutes compared with those who had no siblings. This finding affirms that siblings can discourage VPA in older populations, logically because they spend time with their older siblings in an African environment in sedentary conditions such as sitting idle to chat and viewing TV [5]. As people likely to be in the same age group, older people and their siblings would at best visit each other (in case they live apart) through short walks or motor transport. A short walk taken by an older adult can only lead to moderate PA whereas driving or sitting in a car is a primary sedentary behaviour. Hence, having more siblings is unlikely to encourage VPA in an African sample of older adults. Nevertheless, as our result suggests in line with some studies [6, 10, 19, 21], siblings can contribute to the individual's self-reported health arguably because they can render financial, emotional, and psychological supports that are known in the extant literature to support the health of people who receive them. Based on our result, therefore, we contend that siblings ideally contribute to self-reported health in older populations through social, emotional, and psychological supports and sometimes moderate PA and brief VPA rather than prolonged VPA that lasts not less than 30 minutes per day. This argument throws light on the thinking that prolonged VPA may lead to body pains, fractures, dislocations, and other related casualties.

This study found that older adults who have above 5 other blood relations like cousins and nephews are less likely to report good health compared to those who had no other blood relations, which connotes that having more blood relations who are not siblings and children can be associated with poor self-reported health possibly because these ties increase the economic

burden shouldered by older adults. For instance, older adults may be compelled to spend their income and savings on other blood relations if these kinsmen, who are likely to be aged, suffer from one or more disabilities or/and are unable to provide for themselves. If so, individual older persons may be unable or less able to meet the health care and other personal needs that determine health. This reasoning is consistent with some recent conceptualisations as well as empirical evidence provided in the context of less developed countries such as Ghana, Nigeria, and Pakistan [3, 8, 18]. Even so, some other researchers [17, 19] commented that some blood relations (i.e. aunties, cousins, and nephews) can contribute socially and financially to the health of their kinsmen in a vibrant economic environment where income inequality is low and a high sense of reciprocity abounds. This observation, coupled with the high level of poverty and income inequality in Ghana and other African countries, suggests that blood relations other than children and siblings are unlikely to support the health of their older kinsmen in an environment where socio-economic conditions are poor. That is, blood ties may not have enough resources and the willingness to help older relations in poor countries where people often literally compete for a good life. Even so, the said other blood relations may simply include people with deviant behaviours and, therefore, have a bad influence on older people and their health.

To add, older people who have between 1 and 5 and more than 5 dependents are less likely to engage in prolonged VPA compared with those who have no dependents, which means that dependents do not encourage VPA that lasts 30 or more minutes. Logically speaking, dependents are younger people or individuals who are outside the age group of their older guardians. It thus makes sense to say that older adults may be unwilling to relate with their dependents in ways that result in prolonged VPA. Younger dependents are also more likely to associate with their peers, especially in VPA, but not their older guardians who have different priorities, goals, and social orientations. Dependents could however render other social supports (e.g., running errands and assisting older guardians to access health care and use social amenities) to the benefit of their older guardians. In Ghana, it is quite common to see dependents (including children) accompany their older parents to the hospital. These illustrations are consistent with our result, which indicates that older people who have 1 to 5 or more than 5 dependents are 5 times more likely to report good health compared with their counterparts who do not have any dependents. The empirical literature however presents mixed pieces of evidence that connote that having more dependents can hamper health or may not impact health at all [6, 12, 19].

Older adults who have between 1 and 5 close and mutual friends are less likely to engage in VPA lasting 30 minutes or more as compared with those who have no close and mutual friends. In contrast, older persons who have more than 5 of such friends are 3 times more likely to longer VPA compared with those who do not have any close and mutual friends. Having at least 5 close and mutual friends is thus associated with a higher likelihood of participating in VPA lasting 30 minutes or more, which may be explained by the argument that physical activities are more likely to be initiated and facilitated by close and mutual friends rather than children and other blood relations who are more inclined to relate to their older relatives in sedentary conditions such as viewing TV and chatting while sitting. Our result also indicates that having less than 5 close and mutual friends may discourage VPA; hence older persons need at least five of these peers to maintain or support prolonged VPA. Older people who have at least one close and a mutual friend are however less likely to report having good health, which may be explained by our initial argument that VPA results in body pains and dislocations that may make older individuals frequently feel weak and unhealthy. This thinking resonates with the proposition of ageing biologists that ageing comes with physiological changes (e.g., bone and muscle loss) and senescence that form the basis of deteriorating health in older

populations [22]. They further explained that prolonged VPA can lead to minor or major side effects, including body pains, fractures, and dislocations. Older people who suffer these effects are likely to underscore their health. It could also be that health risk factors (e.g., smoking, alcohol intake, poor access to healthcare, etc.) faced by older people counteract the desired impact of prolonged VPA on health. Interestingly, this result is empirically supported in the older and general populations [9, 12, 13], but its realisation in this study may be because of the covariates captured in the model.

Older adults who have support from more than 5 people are less likely to participate in prolonged VPA because social support might have come from social network members like children and other blood relations who, according to the current study, discourage VPA. In the light of our evidence, therefore, it can be concluded that social support does not necessarily promote VPA as posited by some researchers and theories (i.e., ATA and CTA) and that it can only facilitate PA when it is constituted by specific social network members such as close and mutual friends who initiate and facilitate recreation, work, and other forms of activity. In this vein, the study supports Asiamah's [3] adaption of the ATA and DTA and thus calls for the modification of these theories. Furthermore, having support from at least one social network member increases the chance of having good health, an outcome that backs several studies [23, 24], as well as the SCToH and thus, connotes that social assistance, no matter who provides it, contributes to the health of those receiving it. That being so, the ultimate premise of the SCToH should be that social networks only contribute to the health of its members if there is support in them. This view squares with recent commentaries [3, 25] that assert that cohesion, trust, and reciprocity that form the heart of social support are the actual determinants of health in older and general populations.

Finally, older persons who engaged in prolonged VPA are less likely to report good health compared with their peers who participate in VPA for less than 30 minutes. This result, which opposes most previous studies in the context of the general and older populations [11, 13, 26], endorses the theory of older people sustaining injuries, pains, or dislocations from prolonged VPA, but we cannot rule out the influence of covariates captured in the model on this result. We are as a result inclined to recommend mandatory control for all potential confounders in future studies. This suggestion is made in recognition of the limitations of our cross-sectional design, including our relatively small sample, our reliance on a non-probability sampling method, and our inability to adjust for all possible confounders. It is, therefore, understandable that future researchers would have to apply randomised controlled trials and longitudinal designs to assess the effect of social ties on VPA and health. Collinearity may exist between some of the variables (e.g., dependents and children), so future researchers utilising multiple linear regression analysis are encouraged to assess this potential statistical threat to internal validity. Finally, health and VPA should be assessed using objective measures or at least validated scales to avoid the limitations associated with using subjective categorical measures. This study focused on a sample of retired older adults in Accra who had acquired formal education. Since most older adults in Ghana do not have formal education and live in rural areas [3], our findings may not apply to most older Ghanaians aged 60 years or higher. Despite the foregoing limitations, we believe the current study and its findings have important implications for future research and practice that are discussed in the next section.

## Implications for public health action

This study indicates that having more of some social ties reduces the chance of participating in VPA for at least 30 minutes, whereas having more of some other ties increases the chance of engaging in VPA for 30 minutes or more. This outcome suggests that not all social connections

contribute to VPA and that some social ties discourage it. Depending on the results of future studies, older adults would need to be made aware, through health promotion programs, of the nature of the impact of each domain of their social network on PA. This effort could equip older people with relevant information towards utilising available social network members. For instance, if it is well established empirically that dependents rather discourage VPA, it would be necessary for health promoters to craft and suggest life course strategies by which older people can leverage dependents to maintain VPA, thereby avoiding the adverse impact of living with dependents on VPA. Yet, all social ties are important and relevant to later life because even domains of one's social network that do not directly contribute to VPA can directly or indirectly support health as our results suggest. In other words, social ties that do not contribute to VPA can accord older people other forms of support towards maintaining health. As a result, the need for older people to maintain social ties over time always holds. Similarly, social ties that fail to predict health can directly support VPA or reduce the risk of some diseases (e.g., cardiovascular diseases, neurodegenerative disorders) and, therefore, facilitate healthy ageing. For example, living in the peaceful company of close and mutual friends, which predict self-reported health in this study, could support mental health and slow down the progression of dementia and other neurodegenerative diseases.

As recently reported [5], VPA can be associated with side effects (e.g., body pains, dislocations, fractures, etc.) that can compel older adults to poorly rate their health. This view is particularly applicable to older people who did not grow with the habit of exercising regularly but recently took up VPA as a hobby. Possibly, therefore, regular participation in VPA throughout the lifespan is the ideal life course approach to healthy ageing. More so, learning to undertake VPA regularly as a lifespan habit (that starts at childhood) could be a way to get used to VPA and avoid the said side effects in old age. Thus, individuals must not wait till old age or when they face serious risks of disease to inculcate healthy habits such as VPA. Finally, this study suggests that it may be misleading to relate social ties in the form of a construct to PA and health outcomes. Given the tendency of a construct making a significant association with PA or a health outcome even when some of its domains do not, future researchers would have to relate social network size in terms of its indicators to VPA and health outcomes. This approach would make it possible for quantitative researchers to reach a consensus regarding specific social ties that support VPA and health, thereby creating an opportunity for the ATA, CTA and SCToH to be modified or better aligned with emerging population dynamics.

## Conclusion

Apart from 'other blood relations', all social ties were significantly associated with VPA in that having more children offered older adults a higher chance of participating in VPA lasting 30 or more minutes but having between 1 and 5 siblings had a lower chance of participating in VPA for 30 or more minutes. More so, older adults who had more than 5 siblings were more likely to report good health compared with those who did not have any siblings. Having 1 or more dependents reduced the chance of engaging in VPA for 30 or more minutes a day. This result was consistent with having between 1 and 5 close and mutual friends, but having more than 5 mutual friends offered a higher chance of engaging in VPA for 30 or more minutes. Having one or more people as a source of social support decreased the chance of engaging in VPA for 30 or more minutes but increased the chance of reporting self-reported health. It is concluded that not all social ties support good health in older populations and encourage participation in VPA, but support coming from them is more likely to result in good health. Finally, VPA is significantly associated with self-reported health in the sense that older adults who engaged in VPA for 30 minutes or more in a day were less likely to have good health

compared with their peers who engaged in VPA for less than 30 minutes per day. Hence, when social ties and other personal characteristics are adjusted for, increasing VPA does not necessarily enhance self-reported health.

## Supporting information

**S1 Appendix. Questionnaire.**
(DOC)

**S2 Appendix. Variable categorisation, definition, and coding.**
(DOC)

**S1 Data.**
(SAV)

## Acknowledgments

We thank the management of the Social Security and National Insurance Trust (SSNIT) in Accra Ghana for supporting us to gather data for this study.

## Author Contributions

**Conceptualization:** Nestor Asiamah.

**Data curation:** Nestor Asiamah, Richard Eduafo, Richard Borkey.

**Formal analysis:** Nestor Asiamah.

**Investigation:** Nestor Asiamah.

**Methodology:** Nestor Asiamah, Kyriakos Kouveliotis.

**Project administration:** Richard Eduafo, Richard Borkey.

**Supervision:** Kyriakos Kouveliotis.

**Validation:** Kyriakos Kouveliotis, Richard Eduafo, Richard Borkey.

**Visualization:** Richard Eduafo, Richard Borkey.

**Writing – original draft:** Nestor Asiamah.

**Writing – review & editing:** Nestor Asiamah, Kyriakos Kouveliotis, Richard Eduafo, Richard Borkey.

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
