## [Decision Letter · Decision Letter 0]

22 Mar 2022

PGPH-D-22-00417

Associations between Vigorous Physical Activity, Social Ties, Social Support, and Self-Reported Health among Older Adults in Accra, Ghana

Dear Dr. Asiamah,

Thank you for submitting your manuscript to PLOS Global Public Health. After careful consideration, we feel that it has merit but does not fully meet PLOS Global Public Health’s publication criteria as it currently stands. Therefore, we invite you to submit a revised version of the manuscript that addresses the points raised during the review process.

We look forward to receiving your revised manuscript.

Kind regards,

Guglielmo Campus, Ph.D DDS

Academic Editor

Journal Requirements:

1. We see that your study includes live participants, but you have not included an Ethics Statement. Please update your manuscript file to include an Ethics Statement subsection to your Materials and Methods section. It should include:

i) The full name(s) of the Institutional Review Board(s) or Ethics Committee(s)

ii) The approval number(s), or a statement that approval was granted by the named board(s) 

iii) (for human participants or donors) - A statement that formal consent was obtained (must state whether verbal/written) OR the reason consent was not obtained (e.g. anonymity)

2. Please update the completed 'Competing Interests' statement, including any COIs declared by your co-authors. If you have no competing interests to declare, please state "The authors have declared that no competing interests exist". 

3. Please ensure you include your funder's role statement "The funders had no role in study design, data collection and analysis, decision to publish, or preparation of the manuscript." at the end of your amended statement

4. In the online submission form, you indicated that "The data will be made available upon reasonable request.". All PLOS journals now require all data underlying the findings described in their manuscript to be freely available to other researchers, either 1. In a public repository, 2. Within the manuscript itself, or 3. Uploaded as supplementary information.

5. We do not publish any copyright or trademark symbols that usually accompany proprietary names, eg (R), (C), or TM  (e.g. next to drug or reagent names). Therefore please remove all instances of trademark/copyright symbols throughout the text, including (R) on page 23.

Additional Editor Comments (if provided):

Reviewers' comments:

Reviewer's Responses to Questions

**Comments to the Author**

1. Does this manuscript meet PLOS Global Public Health’s publication criteria? Is the manuscript technically sound, and do the data support the conclusions? The manuscript must describe methodologically and ethically rigorous research with conclusions that are appropriately drawn based on the data presented.

Reviewer #1: Yes

Reviewer #2: Yes

2. Has the statistical analysis been performed appropriately and rigorously?

Reviewer #1: Yes

Reviewer #2: Yes

3. Have the authors made all data underlying the findings in their manuscript fully available (please refer to the Data Availability Statement at the start of the manuscript PDF file)?

Reviewer #1: Yes

Reviewer #2: No

4. Is the manuscript presented in an intelligible fashion and written in standard English?

Reviewer #1: Yes

Reviewer #2: Yes

5. Review Comments to the Author

Reviewer #1: Congratulations to the research team. This study appears to be very thoughtful with sound analytical basis which produced findings which collaborates existing evidence as well as provide new insights into vigorous physical activity and self-reported health impacts among older adults especially relating to social ties and social supports from children and siblings or blood relations. The public health implications as articulated in the conclusion are relevant for the improvement of health and wellbeing relating emanating from physical activity being it moderate or vigorous. Congratulations.

Reviewer #2: This is an interesting study, given that we are moving to an aging population that needs support in terms of improving their health outcome. I have the following comments that need to be considered by the author:

1. It is not clear which institution or organization provided ethical approval for the study.

2. Table 2: Chi square sign is missing, p value = 0.000 should be replaced with p < 0.001

3. Authors claim to have done adjusted Odd Ratios (ORs); however, table 3 does not show results for the adjusted odds ratio

4. Study limitation is missing, given that the population studied are elderly people who accessed Social Security and National Insurance Trust (SSNIT) in Accra, what about others who did not and maybe those who are not covered by the insurance trust.

6. PLOS authors have the option to publish the peer review history of their article (what does this mean?). If published, this will include your full peer review and any attached files.

**Do you want your identity to be public for this peer review?** For information about this choice, including consent withdrawal, please see our Privacy Policy.

Reviewer #1: No

Reviewer #2: **Yes: **Dr. Collins Asweto

---

## [Decision Letter · Decision Letter 1]

7 Jun 2022

PGPH-D-22-00417R1

Associations between Vigorous Physical Activity, Social Ties, Social Support, and Self-Reported Health among Older Adults in Accra, Ghana

Dear Dr. Asiamah,

Thank you for submitting your manuscript to PLOS Global Public Health. After careful consideration, we feel that it has merit but does not fully meet PLOS Global Public Health’s publication criteria as it currently stands. Therefore, we invite you to submit a revised version of the manuscript that addresses the points raised during the review process.

We look forward to receiving your revised manuscript.

Kind regards,

Guglielmo Campus, Ph.D DDS

Academic Editor

Journal Requirements:

Additional Editor Comments (if provided):

There are some work to be done before to move the paper further.

Please check the comments of the reviewer 2.

Reviewers' comments:

Reviewer's Responses to Questions

**Comments to the Author**

1. If the authors have adequately addressed your comments raised in a previous round of review and you feel that this manuscript is now acceptable for publication, you may indicate that here to bypass the “Comments to the Author” section, enter your conflict of interest statement in the “Confidential to Editor” section, and submit your "Accept" recommendation.

Reviewer #3: All comments have been addressed

Reviewer #4: All comments have been addressed

2. Does this manuscript meet PLOS Global Public Health’s publication criteria? Is the manuscript technically sound, and do the data support the conclusions? The manuscript must describe methodologically and ethically rigorous research with conclusions that are appropriately drawn based on the data presented.

Reviewer #3: Yes

Reviewer #4: Yes

3. Has the statistical analysis been performed appropriately and rigorously?

Reviewer #3: Yes

Reviewer #4: Yes

4. Have the authors made all data underlying the findings in their manuscript fully available (please refer to the Data Availability Statement at the start of the manuscript PDF file)?

Reviewer #3: Yes

Reviewer #4: Yes

5. Is the manuscript presented in an intelligible fashion and written in standard English?

Reviewer #3: Yes

Reviewer #4: Yes

6. Review Comments to the Author

Reviewer #3: The article finds association between VPA / Self-reported health with the social ties and social support. While the authors seem to have addressed most of the comments by previous referee, I have some major concerns regarding the work. The authors have begun with the VPA but have not mentioned whether it is in the form of exercise for health promotion or through employment related reasons. If it is the first, then my concerns are not valid. However, if it is the latter or a mix of both – old age employment (in the informal sector in LMICs) cannot be ignored as it might be driven by lack of formal social security or informal social support and social ties. I feel that this line of reasoning is central to this work which has been not emphasised upon until the discussion section. Therefore, this might require a major rewriting of the paper if I understand it correctly. In addition, there are few minor comments and clarification.

(i) People with blindness and mental disorders were excluded through a screening. Please elaborate what screening procedure was used. Kindly include this in the appendix, if appropriate.

(ii) The concept of negative social capital – does it exist or is it a formulation of authors? In any case this require some elaboration with references.

(iii) Consider including the survey tool in the appendix for the readers.

(iv) Social ties and Social support – Please define which variables that have been collected have been used to explain these concepts. with respect to the variables you have collected.

(v) Table 1 – Vigorous PA >= 30 code is missing, kindly update. Moreover, table 1 can be considered moving to the appendix.

(vi) Pg 13, second para – “…. other than siblings and children mar self-reported health …” Does not make sense – possibly there is some typo.

(vii) Please correct “dependant” that has been used throughout the paper.

Reviewer #4: This is a well written Paper. Its was able to within existing literature decribe limitations and addrressed limitation of previous studies in a logical and compelling way.

7. PLOS authors have the option to publish the peer review history of their article (what does this mean?). If published, this will include your full peer review and any attached files.

**Do you want your identity to be public for this peer review?** For information about this choice, including consent withdrawal, please see our Privacy Policy.

Reviewer #3: No

Reviewer #4: **Yes: **Dr Cornelius Ohonsi Ehizokhai

---

## [Decision Letter · Decision Letter 2]

25 Aug 2022

PGPH-D-22-00417R2

Associations between Vigorous Physical Activity, Social Ties, Social Support, and Self-Reported Health among Older Adults in Accra, Ghana

Dear Dr. Asiamah,

Thank you for submitting your manuscript to PLOS Global Public Health. After careful consideration, we feel that it has merit but does not fully meet PLOS Global Public Health’s publication criteria as it currently stands. Therefore, we invite you to submit a revised version of the manuscript that addresses the points raised during the review process.

We look forward to receiving your revised manuscript.

Kind regards,

Guglielmo Campus, Ph.D DDS

Academic Editor

Journal Requirements:

1. We have noticed that you have uploaded Supporting Information files, but you have not included a list of legends. Please add a full list of legends for your Supporting Information files after the references list. 

Additional Editor Comments (if provided):

Reviewers' comments:

Reviewer's Responses to Questions

**Comments to the Author**

1. If the authors have adequately addressed your comments raised in a previous round of review and you feel that this manuscript is now acceptable for publication, you may indicate that here to bypass the “Comments to the Author” section, enter your conflict of interest statement in the “Confidential to Editor” section, and submit your "Accept" recommendation.

Reviewer #5: (No Response)

Reviewer #6: All comments have been addressed

2. Does this manuscript meet PLOS Global Public Health’s publication criteria? Is the manuscript technically sound, and do the data support the conclusions? The manuscript must describe methodologically and ethically rigorous research with conclusions that are appropriately drawn based on the data presented.

Reviewer #5: Partly

Reviewer #6: Yes

3. Has the statistical analysis been performed appropriately and rigorously?

Reviewer #5: Yes

Reviewer #6: Yes

4. Have the authors made all data underlying the findings in their manuscript fully available (please refer to the Data Availability Statement at the start of the manuscript PDF file)?

Reviewer #5: Yes

Reviewer #6: Yes

5. Is the manuscript presented in an intelligible fashion and written in standard English?

Reviewer #5: Yes

Reviewer #6: Yes

6. Review Comments to the Author

Reviewer #5: I commend the persistent effort of the authors with the revision of this article. However, I would like to reiterate the comment of the previous reviewer on the need for clearly describing the key variables as part of the article.

VPA, Social ties, social support etc. are mentioned as key variables in the papers. Since no previously validated tools have been used to assess these variables, a clear description of how these variables have been operationally defined in this study would help the reader with clear understanding.

Moreover, the questionnaire (Appendix A) include a section on health behaviors such as smoking, alcohol consumption, duration of watching TV etc. and type of food consumption. But, these variables are missing in the analysis. Consider including these variables in the analysis as they are probable risk factors of health status.

Reviewer #6: The authors have adequately and exhaustively addressed all the concerns raised by all the reviewers from the first and second review processes.

7. PLOS authors have the option to publish the peer review history of their article (what does this mean?). If published, this will include your full peer review and any attached files.

**Do you want your identity to be public for this peer review?** For information about this choice, including consent withdrawal, please see our Privacy Policy.

Reviewer #5: **Yes: **Deepanjali Behera

Reviewer #6: **Yes: **Ruxton Adebiyi

---

## [Decision Letter · Decision Letter 3]

18 Oct 2022

PGPH-D-22-00417R3

Associations between Vigorous Physical Activity, Social Ties, Social Support, and Self-Reported Health among Older Adults in Accra, Ghana

Dear Dr. Asiamah,

Thank you for submitting your manuscript to PLOS Global Public Health. After careful consideration, we feel that it has merit but does not fully meet PLOS Global Public Health’s publication criteria as it currently stands. Therefore, we invite you to submit a revised version of the manuscript that addresses the points raised during the review process.

We look forward to receiving your revised manuscript.

Kind regards,

Guglielmo Campus, Ph.D DDS

Academic Editor

Journal Requirements:

Additional Editor Comments (if provided):

Reviewers' comments:

Reviewer's Responses to Questions

**Comments to the Author**

1. If the authors have adequately addressed your comments raised in a previous round of review and you feel that this manuscript is now acceptable for publication, you may indicate that here to bypass the “Comments to the Author” section, enter your conflict of interest statement in the “Confidential to Editor” section, and submit your "Accept" recommendation.

Reviewer #6: All comments have been addressed

Reviewer #7: (No Response)

2. Does this manuscript meet PLOS Global Public Health’s publication criteria? Is the manuscript technically sound, and do the data support the conclusions? The manuscript must describe methodologically and ethically rigorous research with conclusions that are appropriately drawn based on the data presented.

Reviewer #6: Yes

Reviewer #7: Yes

3. Has the statistical analysis been performed appropriately and rigorously?

Reviewer #6: I don't know

Reviewer #7: Yes

4. Have the authors made all data underlying the findings in their manuscript fully available (please refer to the Data Availability Statement at the start of the manuscript PDF file)?

Reviewer #6: Yes

Reviewer #7: Yes

5. Is the manuscript presented in an intelligible fashion and written in standard English?

Reviewer #6: Yes

Reviewer #7: No

6. Review Comments to the Author

Reviewer #6: I believe the authors did a good job at addressing the concerns of the reviewers from the previous rounds of reviews.

That said, I have some minor concerns:

1. The statistical method does not mention the criteria for variable selection for the multivariable model.

2. How was collinearity addressed in your adjusted model? I reckon that potential collinearity exists between variables such as dependents vs. children; the number of close friends vs. social support.

3. Did you present the crude odd ratio and adjusted odd ratio in Table 2? It is a bit confusing. I recommend indicating crude odd ratio as 'OR', and Adjusted odd ratio as 'aOR'

4. I could not locate Appendix B.

Reviewer #7: The findings section of the manuscript needs grammatical revisions. The missing of words like "were" is making it difficult to understand the message being communicated.

A revision read to correct any grammatical errors in the manuscript can help improve the written quality of the draft especially for the findings section to make it much easier for the people reading to understand the findings of the research.

7. PLOS authors have the option to publish the peer review history of their article (what does this mean?). If published, this will include your full peer review and any attached files.

**Do you want your identity to be public for this peer review?** For information about this choice, including consent withdrawal, please see our Privacy Policy.

Reviewer #6: **Yes: **Ruxton Adebiyi

Reviewer #7: No

---

## [Decision Letter · Decision Letter 4]

17 Jan 2023

Associations between Vigorous Physical Activity, Social Ties, Social Support, and Self-Reported Health among Older Adults in Accra, Ghana

PGPH-D-22-00417R4

Dear Dr. Asiamah,

We are pleased to inform you that your manuscript 'Associations between Vigorous Physical Activity, Social Ties, Social Support, and Self-Reported Health among Older Adults in Accra, Ghana' has been provisionally accepted for publication in PLOS Global Public Health.

Best regards,

Zulkarnain Jaafar

Academic Editor

Reviewer Comments (if any, and for reference):

Reviewer's Responses to Questions

**Comments to the Author**

1. If the authors have adequately addressed your comments raised in a previous round of review and you feel that this manuscript is now acceptable for publication, you may indicate that here to bypass the “Comments to the Author” section, enter your conflict of interest statement in the “Confidential to Editor” section, and submit your "Accept" recommendation.

Reviewer #7: All comments have been addressed

2. Does this manuscript meet PLOS Global Public Health’s publication criteria? Is the manuscript technically sound, and do the data support the conclusions? The manuscript must describe methodologically and ethically rigorous research with conclusions that are appropriately drawn based on the data presented.

Reviewer #7: Yes

3. Has the statistical analysis been performed appropriately and rigorously?

Reviewer #7: Yes

4. Have the authors made all data underlying the findings in their manuscript fully available (please refer to the Data Availability Statement at the start of the manuscript PDF file)?

Reviewer #7: Yes

5. Is the manuscript presented in an intelligible fashion and written in standard English?

Reviewer #7: Yes

6. Review Comments to the Author

Reviewer #7: The previous comment of the findings section having grammatical errors have been addressed and now is clear to understand upon the first read.

7. PLOS authors have the option to publish the peer review history of their article (what does this mean?). If published, this will include your full peer review and any attached files.

**Do you want your identity to be public for this peer review?** For information about this choice, including consent withdrawal, please see our Privacy Policy.

Reviewer #7: **Yes: **AYESHA ANWAAR CHAUDHRY
